# Antibody-directed evolution reveals a mechanism for enhanced neutralization at the HIV-1 fusion peptide site

Bailey B. Banach[1], Sergei Pletnev[2], Adam S. Olia[2], Kai Xu[2,3], Baoshan Zhang[2], Reda Rawi [2], Tatsiana Bylund[2], Nicole A. Doria-Rose [2], Thuy Duong Nguyen[4], Ahmed S. Fahad[4], Myungjin Lee [2], Bob C. Lin[2], Tracy Liu[2], Mark K. Louder [2], Bharat Madan[4], Krisha McKee[2], Sijy O'Dell[2], Mallika Sastry[2], Arne Schön[5], Natalie Bui[4], Chen-Hsiang Shen[2], Jacy R. Wolfe[4], Gwo-Yu Chuang[2], John R. Mascola[2], Peter D. Kwong [2] ✉ & Brandon J. DeKosky [4,6,7,8] ✉

The HIV-1 fusion peptide (FP) represents a promising vaccine target, but global FP sequence diversity among circulating strains has limited anti-FP antibodies to ~60% neutralization breadth. Here we evolve the FP-targeting antibody VRC34.01 in vitro to enhance FP-neutralization using site saturation mutagenesis and yeast display. Successive rounds of directed evolution by iterative selection of antibodies for binding to resistant HIV-1 strains establish a variant, VRC34.01_mm28, as a best-in-class antibody with 10-fold enhanced potency compared to the template antibody and ~80% breadth on a cross-clade 208-strain neutralization panel. Structural analyses demonstrate that the improved paratope expands the FP binding groove to accommodate diverse FP sequences of different lengths while also recognizing the HIV-1 Env backbone. These data reveal critical antibody features for enhanced neutralization breadth and potency against the FP site of vulnerability and accelerate clinical development of broad HIV-1 FP-targeting vaccines and therapeutics.

A major goal of HIV-1 vaccine development is to elicit broadly neutralizing antibody responses that can provide sterilizing immunity against infection. However, several molecular features constrain neutralizing antibody recognition of the Envelope (Env) trimer and impede in vivo development of broadly neutralizing antibodies (bNAbs). Viral mutation rates produce a highly diverse, unstable, fusion-capable, glycan-shielded Env structure that evades neutralizing antibody recognition. Once precursor B cells are initiated, broadly neutralizing antibody development is still hindered by the limited number of antibody mutations readily sampled in antigen-dependent B cell maturation, paired with the limited HIV-1 diversity that most chronically infected patients are exposed to relative to global HIV-1 diversity. Approximately half of individuals infected with HIV-1 produce bNAbs that neutralize up to ~50% of circulating HIV-1 strains after several years of infection, and highly potent bNAbs have also been discovered in cases of chronic HIV-1 infection onset in fetal development[1,2]. Preliminary investigations using stabilized versions of Env immunogens have elicited neutralizing antibody responses at potencies relevant for protection against natural infection, but these responses were primarily strain- or clade-specific leading to gaps in protection against

[1]Bioengineering Graduate Program, The University of Kansas, Lawrence, KS 66045, USA. [2]Vaccine Research Center, National Institute of Allergy and Infectious Diseases, National Institutes of Health, Bethesda, MD 20814, USA. [3]Department of Veterinary Biosciences, The Ohio State University, Columbus, OH 43210, USA. [4]Department of Pharmaceutical Chemistry, The University of Kansas, Lawrence, KS 66045, USA. [5]Department of Biology, John Hopkins University, Baltimore, MD 21218, USA. [6]Department of Chemical Engineering, The University of Kansas, Lawrence, KS 66045, USA. [7]Department of Chemical Engineering, Massachusetts Institute of Technology, Cambridge, MA 02139, USA. [8]The Ragon Institute of MGH, MIT, and Harvard, Cambridge, MA 02139, USA. ✉e-mail: pkwong@mail.nih.gov; dekosky@mit.edu

globally circulating strains[3,4]. The constraints on natural HIV-1 bNAb development underscore an urgent need to identify critical molecular mechanisms that can be translated into effective broadly neutralizing antibody vaccine strategies.

Based on insights from a panel of well-characterized bNAbs, a handful of neutralizing epitopes on the Env trimer have been identified as targets of rare cross-clade-neutralizing that are candidates for targeted vaccine strategies. Accumulating evidence for targeted vaccine approaches has revealed unique challenges associated with translating each known vulnerable epitope into effective targeted antibody responses. For example, broad and potent recognition of the CD4-binding site requires extensive antibody somatic hypermutation to recognize high viral diversity at this epitope[5–7]. In contrast, the quaternary V1V2 site at the trimer apex requires unusually rare antibody recombination events for potent neutralizing recognition[8–13]. Similarly, neutralization directed against the glycan-V3 supersite requires antibody maturation to recognize an N-linked glycan[14–17], and the proximity of the membrane-proximal external region (MPER) epitope with the viral lipid membrane may necessitate disruption of immune tolerance for effective vaccine-based antibody elicitation[18–25]. The discovery of the bNAb VRC34.01, isolated from a chronically HIV-1-infected donor, identified the fusion peptide (FP) of the HIV-1 Env gp41 subunit as another specific target for broad immune recognition[26]. A majority of VRC34.01's total interactive surface area (55%) binds to solvent-exposed FP residues 512–519 at the N-terminus of gp41, while 26% of the surface co-recognizes glycan N88 on gp120. Trimer-bound VRC34.01 prevents Env subunits from entering into a co-receptor engagement conformation that is required for viral entry, leading to 50% neutralization breadth of 208 globally representative HIV-1 isolates[26].

The biophysical insights defined by VRC34.01's unique FP-recognition mechanisms catalyzed advances in broad HIV-1 vaccine design against the vulnerable FP epitope. An experimental vaccine strategy first priming an antibody immune response using an 8-amino acid FP-based multivalent immunogen (FP8-KLH) followed by multiple Env trimer immunogen boosts produced substantial recognition of the FP epitope and enhanced vaccine-elicited antibody neutralization breadth[27]. The first-generation prime-boost vaccine-elicited FP-mAbs showed modest breadth of up to 31% on a 208 HIV-1 isolate panel[28], and second-generation studies further refined the FP-prime and trimer-boost immunization regimen to produce cross-reactive neutralizing antibodies achieving up to 59% breadth on a 208-strain HIV-1 isolate panel[29]. These preliminary studies demonstrated the potential of FP-specific prime-boost immunization strategies to elicit broad antibody responses on par with natural bNAb development, but at a fraction of the timescale required[29]. However, establishing a truly pan-neutralizing FP-directed epitope vaccine remains an outstanding challenge. Diverse fusion peptide sequences from 3,942 HIV-1 isolates reveal considerable variation among the FP residues[26], and structural analyses of vaccine-elicited anti-FP antibodies revealed that cross-clade neutralization was hampered by sequence-specific FP conformational diversity[30,31]. Identification of a structural and mechanistic pathway to achieve truly broad FP-based protection against circulating HIV-1 strains is urgently needed to accelerate these FP-targeted vaccine designs toward improved protective potency and clinical utility.

Natural infection and vaccine-elicited antibodies have not yet revealed a truly broad anti-FP antibody. However, natural immune responses are often limited by the sampling of rare mutations and inter-clonal competition dynamics in secondary lymphoid organs. In contrast, synthetic mutational generation and screening techniques offer in vitro capabilities to identify neutralizing mechanisms that improve antibody activities[32–34]. Prior investigations by our group and others demonstrated that functional improvements in HIV-1 neutralization can be achieved by precision site-saturation mutagenesis (SSM) and yeast display screening of antibody gene variant libraries to identify synergistic combinations of beneficial mutations with improved antibody potency and/or breadth[33,35–38]. Based on these promising data and the urgent need to improve the neutralization breadth of anti-FP vaccines and antibodies, we used directed evolution in a comprehensive in vitro antibody improvement campaign of the template antibody VRC34.01 in an effort to define molecular mechanisms that could enable broad anti-FP antibody recognition.

We hypothesized that enhanced antibody recognition of neutralization-resistant FP residues presented in the context of the full Env trimer would lead to improvements in HIV-1 neutralization breadth. We experimentally sampled FP-specific antibody maturation pathways for VRC34.01 using precision mutation library generation and directed evolution techniques, defining the genetic, structural, and biophysical antibody features associated with potent and broad HIV-1 FP-neutralization. Analysis of mutant antibody libraries screened en masse using yeast display defined several critical mutations that improved FP-focused binding dynamics and expanded HIV-1 recognition breadth, and can now inspire actionable strategies to improve FP-based vaccine designs. Our results define the molecular mechanisms for achieving ~80% HIV-1 neutralization breadth using a fusion peptide-directed antibody that will accelerate the clinical development of broad vaccines and therapeutics against HIV-1.

## Results

### Precise directed evolution enhanced antibody recognition of diverse FP sequences on HIV-1 SOSIP trimers

We engineered a suite of broadly neutralizing anti-FP HIV-1 antibodies by screening for affinity-enhancing mutations in the VRC34.01 variable region that could provide improved HIV-1 neutralization potency and breadth (Fig. 1). Beginning with the VRC34.01 antibody sequence as template, SSM was used to generate single-mutant DNA libraries comprising all possible 7328 single amino acid substitutions across the VRC34.01 variable heavy and variable light chains (Fig. 1A Panel 2 and Supplemental Table 1)[33–35,39]. Single-mutant variant libraries were cloned into a yeast surface display vector containing a galactose-induced bi-directional promoter, leucine zipper, and a FLAG tag to express, assemble, and monitor Fab libraries expressed on the yeast surface for fluorescence-activated cell sorting (FACS) (Fig. 1B and Supplemental Fig. 1A)[33,34,40,41].

We hypothesized that enhanced recognition of diversified HIV-1 FP residues 512-519 presented in context of the full Env trimer could improve antibody neutralization breadth. Accordingly, Fab-expressing yeast libraries were stained with fluorescently labeled BG505.SOSIP gp160-trimers containing diverse FP sequences from circulating HIV-1 strains: FP- v1 (AVGIGAVF), Thai (AVGIGAMI), and v3 (AIGLGAMF) (Fig. 1B and Supplemental Fig. 1A, B)[26]. Single-mutant libraries for heavy and light chains (VH-SSM and VL-SSM, respectively) were separately enriched over three rounds of FACS to fractionate libraries into high-, medium-, and low-binding affinity phenotypes (Fig. 1B and Supplemental Fig. 1B)[33,34]. Sorted libraries were analyzed by next-generation sequencing (NGS) to reveal the composition of pre- and post-sort libraries and bioinformatically track enriched mutant sequences (Fig. 2A and Supplemental Fig. 1C)[33,34]. Mutations were defined by the substituted amino acid residue according to the VRC34.01 template sequence, with Base 1 corresponding to the start of the Framework 1 portion of the variable region. Several key residue substitutions were identified in NGS data with high-affinity enrichment and selectivity for multiple FP antigens, and were selected for expression as soluble IgG for additional study (Fig. 2A, Supplemental Table 2 single-mutant variants identified by yeast-library screening). We observed that not all sort antigens and libraries showed an equal degree of affinity improvement after library screening (Fig. 1B), and we prioritized the selection of high-affinity enriched antibody mutants from FACS libraries, both with high-affinity binding enrichment after

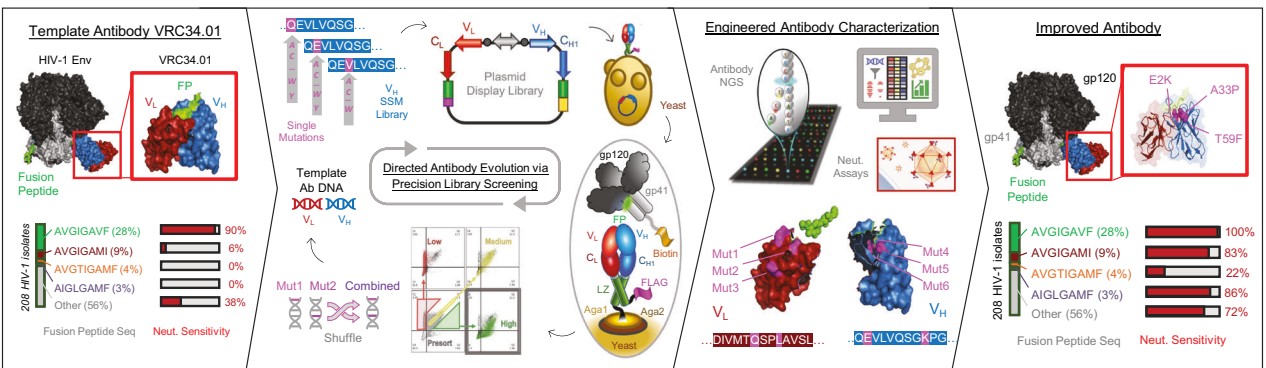

**Fig. 1 | Precision library generation and yeast display screening were applied to the template antibody VRC34.01 to enhance recognition of HIV-1 Envelope (Env)-displayed fusion peptide (FP). A** Workflow for precision anti-HIV-1 antibody engineering via yeast display. The VRC34.01 template antibody variable region genes were mutated and selected for improved HIV-1 affinity via successive rounds of site-saturated mutagenesis (SSM) and DNA shuffling. Antibody mutant libraries were screened using FACS to fractionate mutant populations by antigen binding affinity phenotypes. Next-generation sequencing (NGS) was used to mine antibody sequences and bin antibody variants for trimer binding function based on quantitative variant prevalence analysis across sort groups. Biophysical characterization of engineered single- and multi-mutation antibody variants revealed anti-FP antibody sequence-structure-function relationships and defined potent gain-of-function mechanisms that enhanced FP-targeted HIV-1 neutralization. **B** Yeast libraries expressing antibody in a surface-bound fragment antigen binding (Fab) format were stained with fluorescence markers to measure Fab-surface expression (*Y*-axis) versus antigen binding (*X*-axis) and screened via FACS. Template VRC34.01 (top row) and single-mutant amino-acid substitution libraries were generated via SSM across the entire VRC34.01 variable region heavy (*VH*) and variable region light (*VL*) genes (middle row: pre-sorting populations; bottom row: round 3 high-affinity enriched populations). Libraries are shown bound to diverse FP sequences displayed on HIV-1 trimer probes. Single-mutant libraries were fractionated in three sequential rounds into high-, medium-, and low-affinity performance bins using FACS, resulting in sorted libraries with phenotypically observable differences in trimer binding. Medium-, and low-affinity plots are also provided in Supplemental Fig. 1A.

screening and also with low bioinformatic enrichment ratios (<10) in medium- and low-affinity sorts (Fig. 1B and Supplemental Fig. 1C).

## Numerous single mutations increased antibody neutralization breadth and potency

47 single mutant variants were selected for expression and characterization as IgG protein (Supplemental Table 2; Kabat numbering

also provided for each mutation). Initial characterization included a limited 8-virus neutralization panel designed to identify the most promising single-mutation antibody candidates (Supplemental Fig. 2A and Supplemental Table 3). The eight HIV-1 strain panel was designed so that neutralization breadth would correlate with antibody breadth for other known anti-FP antibodies on a larger 208-virus panel. Panel design was completed using a $10^7$ randomized

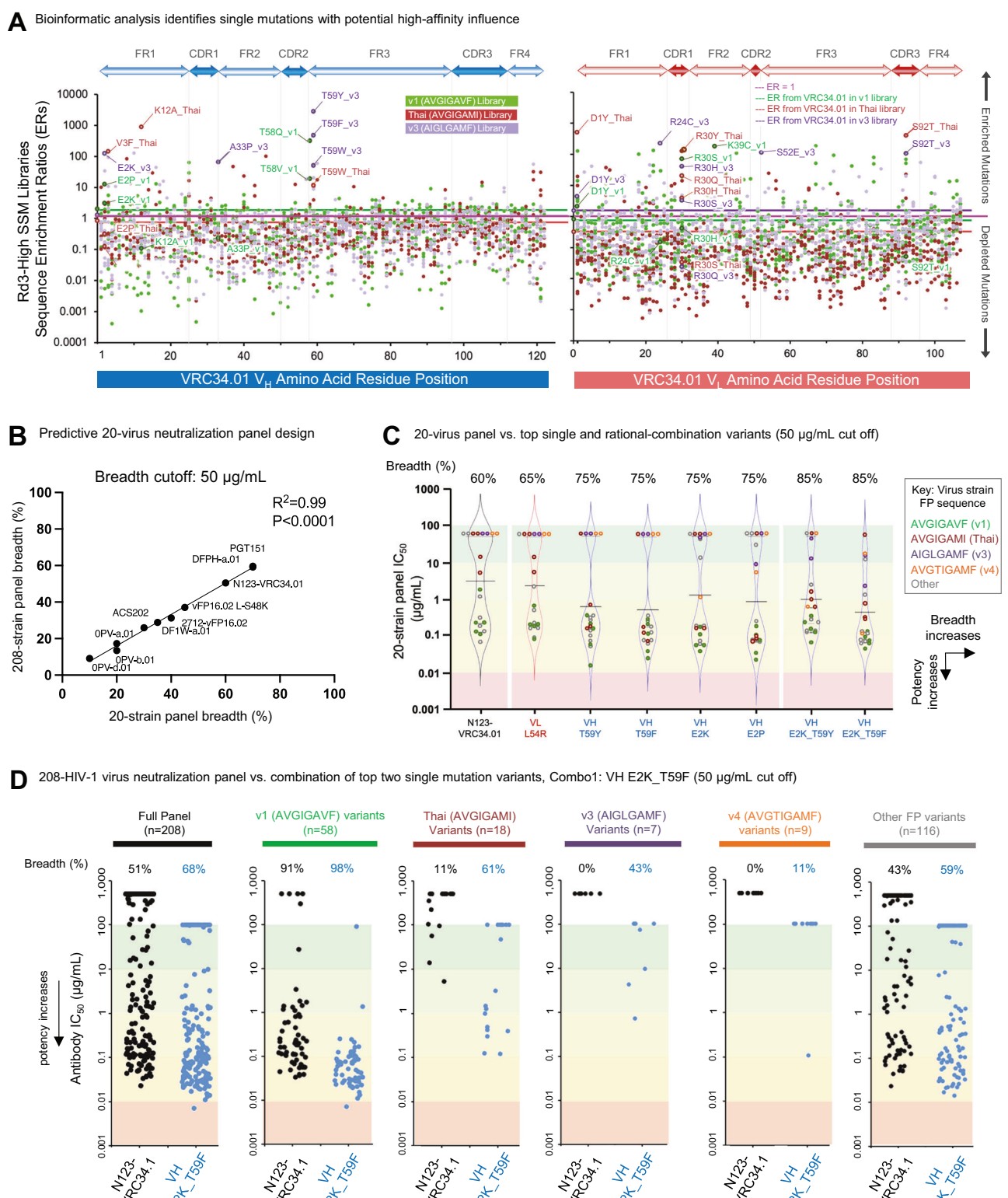

**A** Bioinformatic analysis identifies single mutations with potential high-affinity influence

**B** Predictive 20-virus neutralization panel design

**C** 20-virus panel vs. top single and rational-combination variants (50 μg/mL cut off)

**D** 208-HIV-1 virus neutralization panel vs. combination of top two single mutation variants, Combo1: VH E2K_T59F (50 μg/mL cut off)

search for an 8-virus sub-group within the larger 208-panel that maximized the correlation between select control anti-FP antibody's neutralization outcomes (Supplemental Fig. 2A and Supplemental Table 3). Functional evaluation of the top 47-single mutant variants on the initial predictive 8-virus panel revealed that many mutations had no effect or a detrimental effect on neutralization, however, eight of the top single-mutant antibody variants showed important potency improvements (Supplemental Fig. 2B and Supplemental Table 4).

One variant (VH_E2P) contained a glutamic acid (E) to proline (P) substitution at the second residue in the variable heavy chain and was enriched in yeast library screens against FP-v3 (AIGLGAMF) (Fig. 2A). VH_E2P showed both enhanced HIV-1 neutralization potency and breadth on the 8-virus panel (Supplemental Fig. 2B), achieving gain-of-function neutralization against strain 6405.v4.c34 which was previously resistant to the VRC34.01 template antibody. Notably, strain 6405.v4.c34 displays the same fusion peptide sequence v3 (AIGL-GAMF) that was used as a probe to discover VH_E2P in single-mutant

**Fig. 2 | Bioinformatic mining of single-mutation NGS data from SSM library screens revealed multiple mutations that provided enhanced HIV-1 neutralization potency and breadth. A** Enrichment ratios (ER) are plotted for single mutant antibody sequences derived from Round 3 high-affinity sorted libraries. Mutations were defined by determining the percent identity match to the template gene and denoting the substituted amino acid residue relative to the template sequence, with Base 1 corresponding to the start of the antibody variable region. NGS analysis of single mutant library screens highlighted multiple amino acid substitutions across VRC34.01 antibody variable regions that could enhance diverse HIV-1 FP recognition. **B** A 20-virus panel was optimized for predictive correlation with neutralization breadth on a larger 208-virus panel as a pre-screening tool to down-select promising candidates prior to 208-virus panel neutralization analysis. Template antibody VRC34.01 recognized 60% of strains in the 20-virus panel and ~50% in the 208-virus panel. PGT151, the broadest previously reported antibody that interacts with FP, recognized 70% of strains in the 20-virus panel and

~60% of the broader 208 strains. One-sided p-value for significance of data points deviating from a straight line is shown. No adjustments were made for multiple comparisons. **C** Four single mutations discovered by SSM screening (VH_T59Y, VH_T59F, VH_E2K, VH_E2P) were expressed as soluble IgG and revealed neutralization to 75% of the 20-virus panel. Rational combinations of these top single-mutant variants further improved neutralization breadth, with a maximum 85% breadth on the 20-virus panel achieved for VH_E2K_T59F (also referred to as VRC34.01-Combo1, Kabat numbering VH_E2K_T59F). Antibodies were considered neutralizing if their IC50 potency was <50 μg/mL. **D** 208-virus panel data revealed enhanced breadth for Combo1 (VH_E2K_T59F) to 68% of all strains, with strong neutralization of FP_v1 strains (98%), moderate neutralization of FP_Thai strains (61%), and gain-of-function for neutralization of FP_v3 and FP_v4 strains that were not neutralized at all by VRC34.01 (43% and 11%, respectively). Combo1 showed improvements compared to the template VRC34.01 antibody against all HIV-1 FP subclasses.

yeast library screens. These data suggested that the VH_E2P substitution was functionally enhanced for antigen recognition and may be an important residue for viral neutralization.

A second pass 20-virus neutralization assay was designed using the same randomized search method as in the previous 8-virus panel to further assess top single- and rational-combination mutant antibody variants (Fig. 2B, Supplemental Fig. 2C, and Supplemental Tables 2 and 5; VRC34.01_combo3 and VRC34.01_combo4 did not express). 20-virus panel results demonstrated that a handful of single mutations enriched during yeast library screening could increase neutralization breadth from 60% (template mAb) up to 75% (Fig. 2C VH_T59Y, VH_T59F, VH_E2K, and VH_E2P, Supplemental Fig. 2D), whereas rational combinations of the top single mutations increased breadth further up to 85% (VH_E2K_T59Y and VH_E2K_T59F) with a ~10-fold increase in overall potency for the most improved combination variant, VH_E2K_T59F (Kabat numbering: VH_E2K_T58F), referred to as VRC34.01_combo1 or combo1 (Fig. 2C, Supplemental Fig. 2D, Supplemental Table 2, and Supplemental Data 1).

## Rational combinations of top single mutations improved neutralization performance yet revealed gaps for antibody recognition of circulating HIV-1 strains

VRC34.01_combo1 was next evaluated on a larger 208-virus panel to characterize neutralization breadth against a globally representative panel of HIV-1 and FP sequence diversity (Fig. 2D and Supplemental Data 2). Combo1 comprised two heavy chain substitutions to template N123-VRC34.01 antibody (Fig. 2A and Supplemental Table 2) and showed neutralization breadth improved from 52% up to 70% on the 208-virus panel (Fig. 2D). As shown in Fig. 2D, we found that enhanced breadth was not evenly distributed amongst HIV-1 variants when grouped by FP sequence. Combo1 neutralized FP_v1 (AVGI-GAVF) strains similarly to template antibody VRC34.01 with some potency improvements, and it showed substantially improved neutralization breadth against HIV-1 variants encoding the FP_Thai (AVGIGAMI, improving to 61% breadth from 11% with the template mAb). Combo1 also showed gain-of-function neutralization against 43% of VRC34.01-resistant strains encoding FP_v3 (AIGLGAMF), and also acquired some neutralization gain of function against FP variants not included in single-mutation FACS screening such as FP_v4 (AVGTIGAMF, 0% to 11%) and other rare FP-variants (43–59%, Fig. 2D and Supplemental Data 2). However, despite starting from 0% of FP_v4 strains neutralized by VRC34.01, the improved Combo1 antibody still only neutralized 1 out of 9 FP_v4 strains on the 208-strain panel. As FP_v4 is a 9-mer, compared to most other FP sequences being 8-mers, we hypothesized that further rounds of screening to specifically improve antibody recognition of FP_v4 could further enhance neutralization capacity against HIV-1 strains like FP_v4 that encode longer FP sequences.

## Structural analysis of VRC34.01-combo1 depicts inter-chain interactions with favorable binding dynamics that improve HIV-1 neutralization

To determine the structural basis of increased breadth from Combo1's E2K and T59F mutations, we determined the co-crystal structure of Combo1 Fab in complex with fusion peptide FP8_v1 (AVGIGAVF, Fig. 3A and Supplemental Table 6) and the cryo-EM structure of Combo1 Fab with the BG505 DS-SOSIP trimer (Fig. 3B, Supplemental Fig. 3 and Supplemental Table 7). Cryo-EM mapping revealed similar overall structural interactions that were previously observed for the template VRC34.01 antibody bound to the HIV-1 trimer (Fig. 3B, PDB: 6NC3). We also compared crystal structures of the Combo1:FP8v1 complex (Fig. 3A) to the template VRC34.01:FP8v1 complex (PDB: 5I8C) and observed that the Combo1 structure increased affinity by removing electrostatic conflicts via the VH_E2K substitution, while the VH_T59F substitution also strengthened the hydrogen bond to FP residue G514 (Fig. 3C–E). More specifically, the Combo1 $K2_{HC}$ mutation contributed to increased HIV trimer affinity by removing electrostatic repulsion between template $E2_{HC}$ and $E87_{gp120}$ (Fig. 3C). Super-imposed crystal structures of the Combo1 Fab and VRC34.01 Fab, both in complex with FP, showed how the $F59_{HC}$ substitution enhanced inter-chain hydrophobic interactions to $W50_{HC}$ and $Y94_{HC}$, while also removing unfavorable interactions between the template $T59_{HC}$ hydroxyl group and I515 on the HIV-1 FP (Fig. 3D). The replacement of $T59_{HC}$ with bulky F pushed the side chain of $Y94_{LC}$ closer to FP, enabling the formation of a tighter H-bond with $G514_{FP}$ (Fig. 3D, 3.2 Å versus 3.6 Å). The representative electron density around critical residues $W50_{HC}$, $F59_{HC}$, $Y94_{LC}$, $V513_{FP}$, $G514_{FP}$, and $I515_{FP}$ further suggested that Combo1 enhances FP affinity by supporting favorable electrostatic interactions with HIV-1 trimer (Fig. 3E).

## Screening multi-mutation antibodies against diverse FP sequences of different lengths achieves best-in-class HIV-1 neutralization breadth

Following single mutant library screening and the rational combinations leading to Combo1, we next undertook subsequent rounds of multi-mutation library generation and repeated the screening process against HIV-1 BG505 SOSIP trimers that encoded diverse FP sequences in an effort to address the gaps in neutralization observed for Combo1 against FP_v4 and other diverse HIV-1 FP variants (Figs. 1A and 4A and Supplemental Table 1). Multi-mutation libraries were generated by pooling enriched antibody libraries and performing DNA shuffling to combine the most promising single mutations together for follow-up screening. In addition to DNA shuffling, SSM was also repeated on the shuffled genes to thoroughly screen for potentially synergistic multi-mutation combinations (see Methods). In total, five multi-mutation libraries were generated for yeast display and enriched via FACS against four antigen probes: FP_v1 (AVGIGAVF), Thai (AVGIGAMI), v3

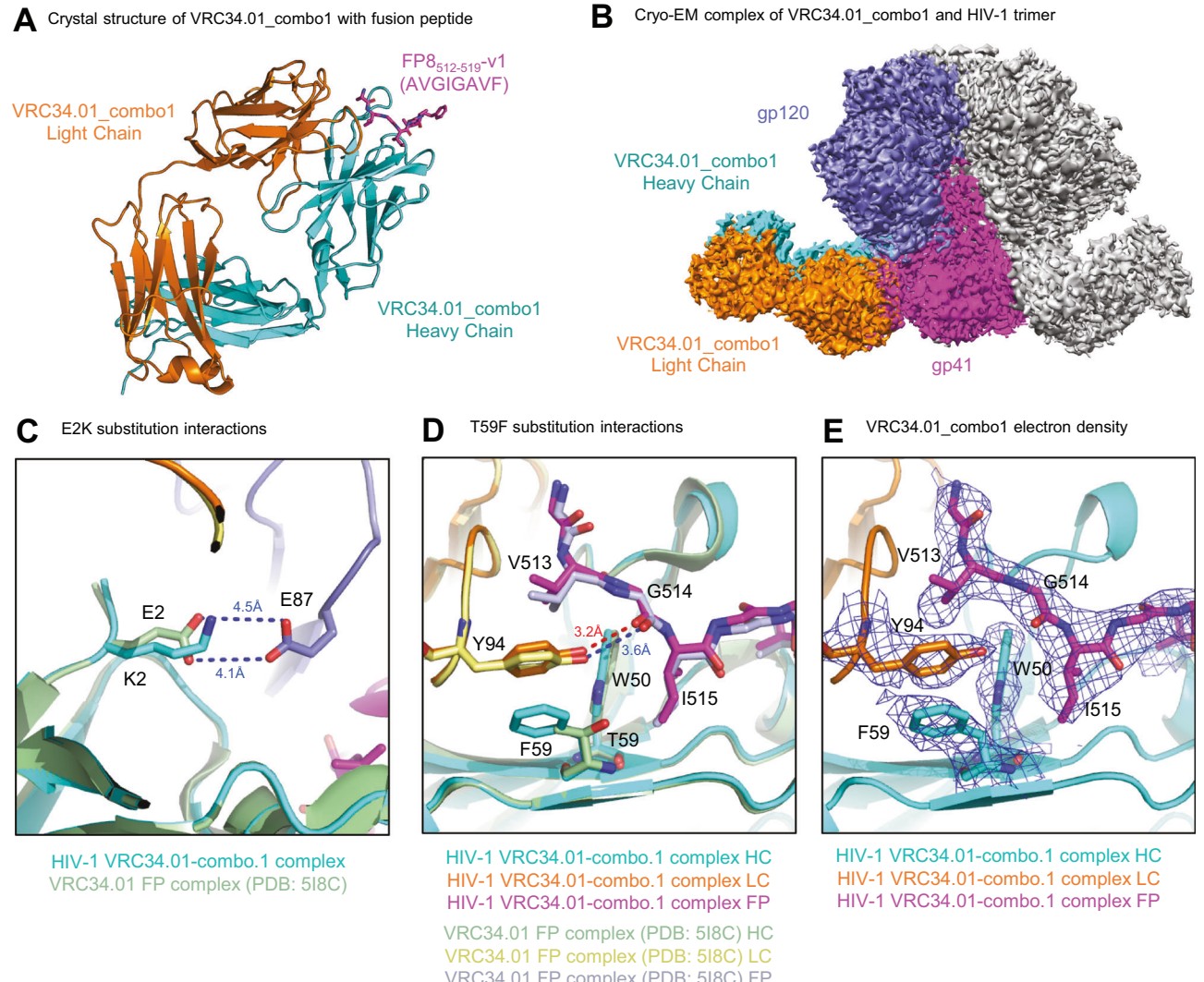

**A** Crystal structure of VRC34.01_combo1 with fusion peptide

VRC34.01_combo1 Light Chain

FP8$_{512-519}$-v1 (AVGIGAVF)

VRC34.01_combo1 Heavy Chain

**B** Cryo-EM complex of VRC34.01_combo1 and HIV-1 trimer

gp120

VRC34.01_combo1 Heavy Chain

VRC34.01_combo1 Light Chain

gp41

**C** E2K substitution interactions

E2
K2
E87
4.5Å
4.1Å

HIV-1 VRC34.01-combo.1 complex
VRC34.01 FP complex (PDB: 5I8C)

**D** T59F substitution interactions

V513
G514
Y94
3.2Å
3.6Å
W50
I515
F59
T59

HIV-1 VRC34.01-combo.1 complex HC
HIV-1 VRC34.01-combo.1 complex LC
HIV-1 VRC34.01-combo.1 complex FP
VRC34.01 FP complex (PDB: 5I8C) HC
VRC34.01 FP complex (PDB: 5I8C) LC
VRC34.01 FP complex (PDB: 5I8C) FP

**E** VRC34.01_combo1 electron density

V513
G514
Y94
W50
I515
F59

HIV-1 VRC34.01-combo.1 complex HC
HIV-1 VRC34.01-combo.1 complex LC
HIV-1 VRC34.01-combo.1 complex FP

**Fig. 3 | Structural basis for VRC34.01-Combo1's recognition of diverse FP.**
**A** Crystal structure of VRC34.01-Combo1 Fab in complex with fusion peptide variant FP8_v1. **B** CryoEM structure of BG505 DS-SOSIP HIV trimer in complex with the VRC34.01_Combo1 Fab. **C, D** Details of interactions between VRC34.01-Combo1 (cyan/orange/magenta) bound to peptide FP8v1 (AVGIGAVF) that provided improved neutralization breadth compared to parental VRC34.01 (green/yellow/violet). **E** Representative electron density around critical residues W50$_{HC}$, F58$_{HC}$, Y94$_{LC}$, V513$_{FP}$, G514$_{FP}$, and I515$_{FP}$.

(AIGLGAMF), and additionally v4 (AVGTIGAMF) in an effort to address the aforementioned gaps in neutralization breadth (Fig. 4A and Supplemental Table 1).

Multi-mutation libraries were enriched for three rounds of FACS, and showed enhanced trimer recognition against all four antigens. In particular, these FACS-enriched libraries showed dramatically enhanced recognition of FP sequences v3 and v4, and modest enhancement against v1 and Thai compared to template VRC34.01 (Fig. 4A). Potentially beneficial combinations of mutations were bioinformatically identified by NGS of heavy and light chain libraries (Fig. 4B). Based on these screens the top 28 rational multi-mutant (mm) combinations were inferred from FACS and NGS performance data (Fig. 4B and Supplemental Table 2). Given the known gaps in VRC34.01 and Combo1 neutralization, emphasis was placed on mutations that enriched against FP8_v4, including VH_A33P and VH_A33P_T59F (Fig. 4B). Soluble IgG expression and characterization of these multi-mutant variants on the 20-virus panel revealed that 10 of the 28 multi-mutant variants achieved a neutralization breadth of 90% or more in the 20-virus panel (Fig. 4C and Supplemental Data 1), and the most improved multi-mutation variant VRC34.01_mm28, or mm28,

used a total of three heavy chain substitutions from the template VRC34.01 to neutralize 100% of the 20 viruses in the panel (Fig. 4C, VH_E2K_A33P_T59F, or in Kabat numbering: VH_E2K_A33P_T58F, Supplemental Table 2). Based on extrapolation beyond known data points using the 20-virus panel model's correlations, VRC34.01_mm28 was thus anticipated to have an anti-FP antibody best-in-class neutralization breadth around 80% (Supplemental Fig. 4A).

Based on the promising 20-virus panel data, we next performed a more comprehensive analysis of HIV-1 neutralization against globally representative FP diversity, revealing improved neutralization breadth to 79% on the 208-virus panel (Fig. 5A and Supplemental Data 3). VRC34.01_Combo1 and _mm28 both showed more potent, broader cross-clade recognition compared to template VRC34.01 (Fig. 5B, C). IC50 neutralization breadth (cutoff: <50 μg/mL) was 50% for VRC34.01, compared to VRC34.01_Combo1 at 68% (20-virus panel prediction: 72% ± 5%) and VRC34.01_mm28 at 79% (20-virus panel prediction: 80% ± 5%, Supplemental Fig. 4A). IC80 neutralization breadth (cutoff: <50 μg/mL) was 23% for VRC34.01, whereas VRC34.01_Combo1 maintained 46% breadth, and VRC34.01_mm28 reached 55% IC80 breadth (Fig. 5C).

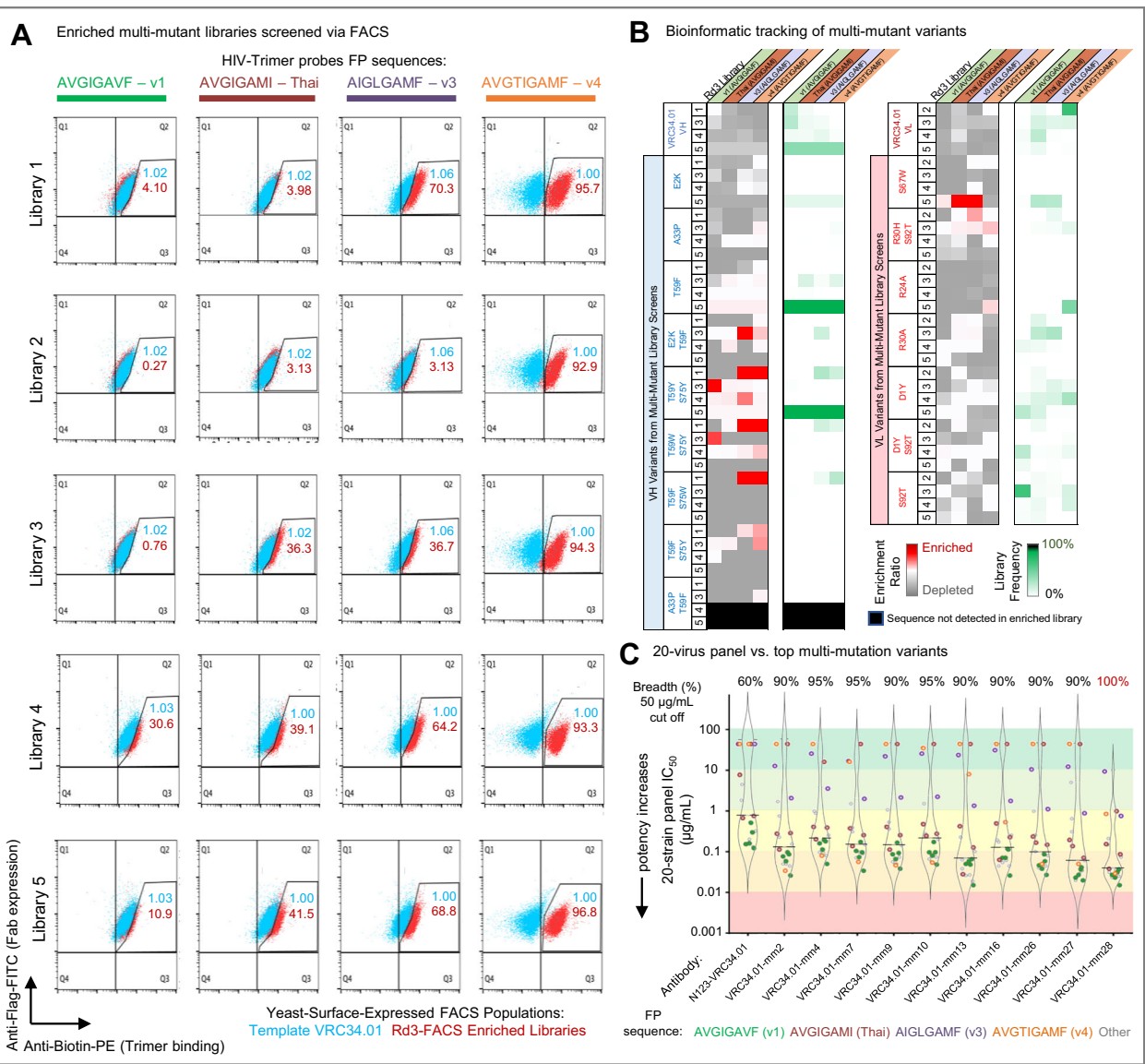

**Fig. 4 | Targeted multi-mutation screening further enhanced neutralization of FP8_v3 and FP8_v4 strains, enabling best-in-class anti-FP neutralization breadth and potency.** A Multi-mutation yeast libraries expressing antibody as surface-bound Fab were stained with fluorescence markers to measure Fab-surface expression (*Y*-axis) versus antigen binding (*X*-axis) and screened via FACS. Multi-mutant yeast display libraries (red) were compared to surface-expressed VRC34.01 template Fab (blue). Multi-mutant library generation and FACS screening provided phenotypically apparent affinity improvements against BG505 SOSIP trimers encoding FP_Thai, FP_v3, and FP_v4 relative to template mAb VRC34.01, with the goal of further improving HIV-1 neutralization by Combo1 (VH_E2K_T59F, Fig. 2D). **B** Heat maps of enriched mutant sequences from screening multi-mutation libraries 1, 2, 3, 4, and 5 against trimer probes displaying four different FP variants. Bioinformatic mining of multi-mutation yeast display library screening data revealed several mutations enriched against diverse FP sequences. **C** When expressed as soluble IgG, numerous multi-mutation variants achieved 95% neutralization in the 20-virus panel, with one variant (VRC34.01_mm28, or mm28, VH_E2K_A33P_T59F) achieving 100% neutralization of HIV-1 strains in the panel. Antibodies were considered neutralizing with an IC50 potency <50 µg/mL.

Like its predecessor VRC34.01_Combo1, multi-mutation variant VRC34.01_mm28's enhanced breadth was not evenly distributed amongst HIV-1 variants when grouped by FP sequence (Fig. 5A, D). Both VRC34.01_Combo1 and VRC34.01_mm28 effectively neutralized strains containing FP_v1 (AVGIGAVF), whereas VRC34.01_mm28 showed more substantially improved neutralization breadth and gain of functions against strains encoding more diverse sequences screened in FACS analysis: FP_Thai (AVGIGAMI), FP_v3 (AIGLGAMF), FP_v4 (AVGTIGAMF) (Figs. 1 and 4), as well as additional rare FP variants not screened in this antibody improvement campaign. Viral neutralization fingerprint analysis confirmed that Combo1 and mm28 still clustered with template VRC34.01 among other FP-specific antibody sequences (Supplemental Fig. 4C).

## Structural analysis of VRC34.01_mm28 revealed superior FP-based neutralization by recognition of both HIV-1 gp160 FP and gp41

We determined cryo-EM and crystal structures of the improved antibody bound to BG505 DS-SOSIP trimer and to FP8v4 peptide, respectively (AVGTIGAMF, Fig. 6 and Supplemental Figs. 5–6 and Supplemental Tables 6 and 7). Structural comparison of mm28, Combo1, and their template VRC34.01 revealed three synergistic interactions that enabled mm28 to achieve enhanced breadth, including the accommodation of insertions in FP (Fig. 6). The cryoEM structure showed nearly identical binding of VRC34_mm28 and VRC34.01 (Fig. 6A, B and Supplemental Fig. 6), relying on recognition of both gp120 and g41 to achieve neutralization. Comparison of

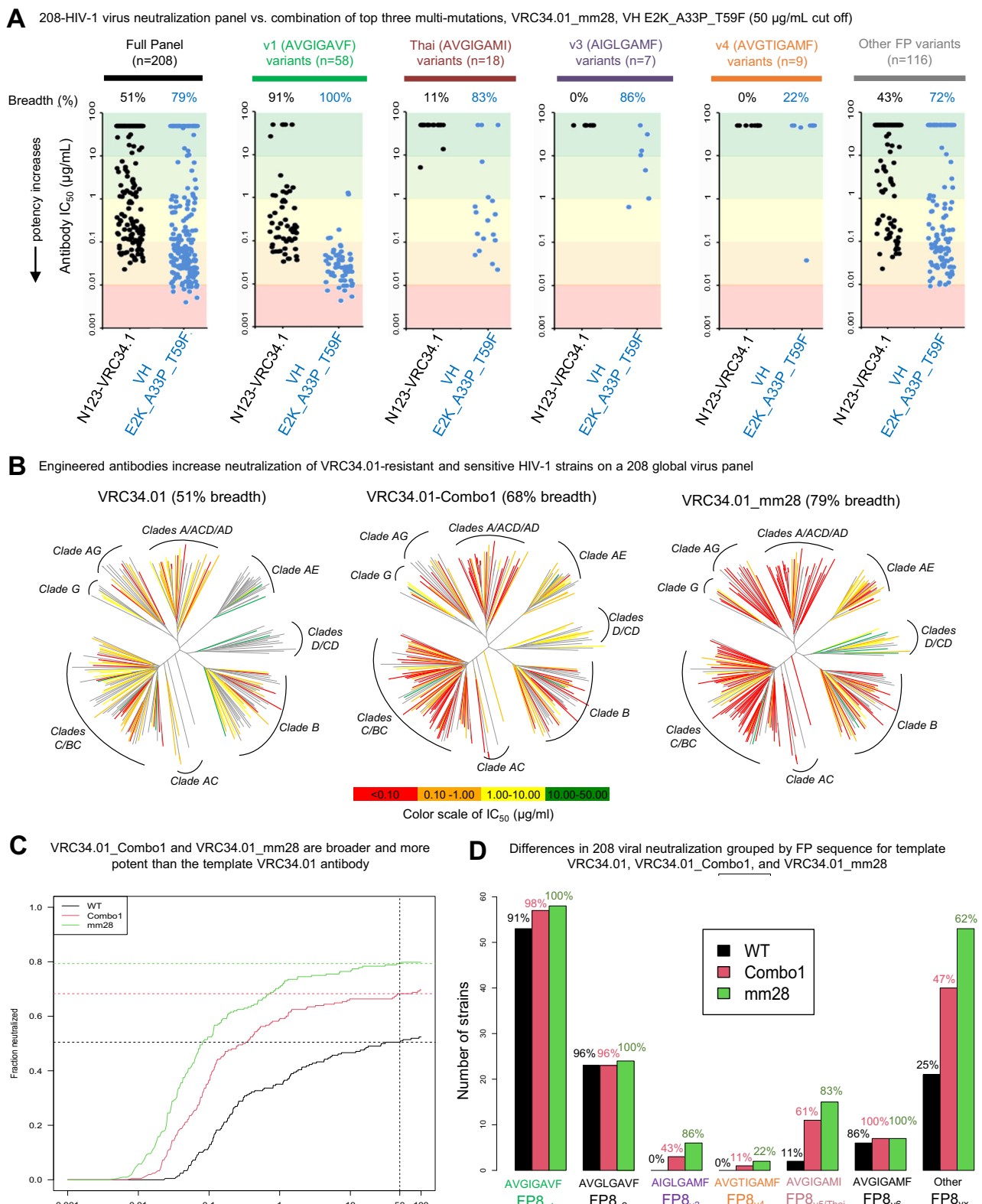

**A** 208-HIV-1 virus neutralization panel vs. combination of top three multi-mutations, VRC34.01_mm28, VH E2K_A33P_T59F (50 µg/mL cut off)

**B** Engineered antibodies increase neutralization of VRC34.01-resistant and sensitive HIV-1 strains on a 208 global virus panel

**C** VRC34.01_Combo1 and VRC34.01_mm28 are broader and more potent than the template VRC34.01 antibody

**D** Differences in 208 viral neutralization grouped by FP sequence for template VRC34.01, VRC34.01_Combo1, and VRC34.01_mm28

epitope topography between template VRC34.01 and a previously characterized germline intermediate: VRC34 I3 (H33P_HC, PDB: 6UCE), revealed that the P33_HC substitution widened the hydrophobic binding groove in the antibody epitope to allow broader recognition of FP sequences with insertions, which included the 9-mer FP_v4 (AVGTIGAMF).

The crystal structure of VRC34.01_mm28 bound to the FP_v4 peptide (Fig. 6C–F) further demonstrated a slight widening of the binding groove in the epitope around the A33P substitution like observed in the intermediate I3 structure (PDB: 6UCE); the A33P_HC mutation in VRC34·mm28 displaced Tyr97_HC by 1.4 Å, which in turn shifted Asn100_HC out and back by 2.5 Å, resulting in an overall

**Fig. 5 | Top VRC34.01 mutant variant characterization out performs template antibody via stepwise improvements in potency and breadth via engineered mutations. A** 208-virus panel neutralization data showed enhanced breadth compared with the template VRC34.01 antibody against all HIV-1 FP subclasses for antibody variant mm28 (VH_E2K_A33P_T59F). When binning neutralization by FP sequences screened in yeast display, mm28 showed 100% neutralization breadth of FP_v1 strains (vs. 91% breadth for the VRC34.01 template antibody), 83% neutralization of FP_Thai strains (vs. 11%), 86% neutralization of FP_v3 (vs. 0%), 22% neutralization FP_v4 strains, (vs. 0%), and 72% neutralization of other diverse FP sequences (vs. 43%). **B** Dendrograms reveal that the engineered antibodies Combo1 (68% total breadth) and mm28 (79% total breath) increased both cross-clade recognition and neutralization against VRC34.01-resistant and sensitive strains. **C** Titration curves show that Combo1 and mm28 gained stepwise improvements in neutralization potency and breadth compared to template VRC34.01. **D** Recognition of VRC34.01-resistant and sensitive HIV-1 strains displaying diverse FP sequences were compared for several antibodies in the study.

broadening of the peptide binding cleft by approximately 1 Å (Fig. 6E, F). Moreover, the larger bulk of the mutant proline side chain at position 33 in mm28 resulted in the Fab binding more surface area at the FP interface: 12.96 Å$^2$ in mm28 as compared to 2.93 Å$^2$ in Combo1.

### Engineered antibodies' biophysical and genetic correlates define mechanisms for broad HIV-1 fusion peptide recognition

We evaluated the sequence-structure-function relationship of improved antibodies to determine biophysical antibody features associated with potent and broad HIV-1 FP-directed neutralization (Fig. 7). Gene-specific substitution profiles (GSSPs) for the VRC34.01 template heavy chain *IGHV1-2* gene defined the rarity of mutations in public human antibody repertoires that could enhance recognition of HIV-1 FP and trimer (Fig. 7A, $n = 108$ donors, 19,143 clones)[42]. Substitutions identified in yeast display are highlighted with cyan background, and a substitution frequency lower than 0.5% was defined as rare mutation and marked by a red square. The *CDR1* and *CDR2* were shown with green and blue boxes, respectively. The most improved antibody mm28 required 4 rare substitutions to the germline variable heavy chain gene that are observed infrequently in human repertoires: V2K, Q3V, Y33A, and T68Y, whereas the template VRC34.01 antibody comprised only two rare mutations.

To define FP sequence requirements for neutralization by Combo1 and mm28, we created a peptide panel comprising single Ala and Gly mutations to the N terminus of FP_v1 and screened the optimized antibodies, along with template VRC34.01, for recognition of Ala-Gly peptide variants (Fig. 7B). Ala–Gly mutations affected recognition by all three antibodies when the mutations occurred within the N-terminal residues of FP (512–516). All antibodies exhibited sensitivity to changes at position 513, indicating requirement of a hydrophobic sidechain for antibody binding. VRC34.01 recognition was very sensitive to changes at FP residues 512$_A$, 513$_V$, 515$_I$, and 516$_G$, and was partially sensitive to changes at 518$_V$ and 519$_F$. Like VRC34.01, the improved antibodies VRC34_Combo1 and mm28 binding sensitivities were also affected by Ala–Gly alterations at residues 513$_V$, 515$_I$, and 516$_G$. However unlike VRC34.01, Combo1 and mm28 recognition was comparably less sensitive to changes at 512$_A$, 518$_V$, and 519$_F$. The most improved antibody, mm28, also exhibited an enhanced ability to tolerate a change at residues 518$_V$ and 519$_F$ and, to a lesser extent, at 515$_I$ and 516$_G$. These results outlined the FP-targeting antibody performance for binding N-terminal residue substitutions, and suggested that the improved accommodation of variable residues in the FP C-terminus enables effective neutralization by mm28.

We next used isothermal calorimetry to measure binding interactions to diverse FP probes and determine the energetic changes from each mutation and mutant combinations that supported enhanced antibody performance (Fig. 7C and Supplemental Table 8 note: the true $K_D$ values could not be obtained by ITC). Six different FP sequences were assessed, including four sequences used in yeast display screening: v1 (AVGIGAVF), Thai (AVGIGAMI), v3 (AIGLGAMF), and v4 (AVGTIGAMF); and two additional sequences from globally circulating HIV-1 strains: v2 (AVGLGAVF) and v3F (AIGLGAVF). The template antibody VRC34.01 bound with the highest affinity to v1 (AVGIGAVF, $K_D = 33$ nM) and v2 (AVGLGAVF, $K_D = 37$ nM), with favorable enthalpy contributions partially opposed by unfavorable entropy changes. The binding of VRC34.01 to the other peptides was significantly weaker,

especially to v3 (AIGLGAMF, $K_D = 2,500$ nM) and Thai, (AVGIGAMI, $K_D = 580$ nM) due to less favorable enthalpy changes in the bound state compared to the decreases in unfavorable entropy.

In comparison, the binding behavior of the single-mutant VH_E2K showed only minor changes in energetics to v1, v2, and Thai resulting in essentially the same or slightly improved affinities compared to VRC34.01, whereas binding to v3F was 50% weaker. However, VH_E2K bound significantly better than VRC34.01 to v3 and v4, with 1.8 and 1.4-fold improved affinity, respectively. The binding behavior of single-mutant VH_T59F revealed affinities with the same or larger entropy penalties but with even larger favorable enthalpy changes than VRC34.01, which resulted in improved binding affinity to all six peptides. The binding affinity of VH_T59F to v2 and v3F were ten and twelve-fold higher, respectively, and over six-fold higher to v4 and Thai, and four-fold higher to v1 and v3. Significantly better binding affinities were observed when VH_E2K and VH_T59F improved single mutations were expressed together in Combo1. Combo1 showed increased affinity for v3 ($K_D = 220$ nM compared to 2500 nM for VRC34.01, 1400 nM for VH_E2K, and 540 nM for VH_T59F) because the gain in favorable enthalpy contribution was larger than the increase in entropy penalty. Combo1 also showed six and twelve-fold better binding to v1 and v3F, respectively, compared to VRC34.01. The top neutralizing multi-mutation variant mm28 (comprised of the heavy chain mutations: E2K, A33P, and T59F) showed the largest favorable enthalpy increases and unfavorable entropy contributions overall, but because the favorable enthalpy increased more than the entropy penalty mm28 bound with the best affinity to all peptides. The binding affinity to FP_V4 was the highest for mm28 compared to the other antibodies, but the magnitudes of the favorable enthalpy and unfavorable entropy changes were much smaller in this case.

To determine the contributions of mutant residue pairings to the overall structural stabilization, residue pair energy analysis was performed using molecular dynamic simulations (Fig. 7D). Residue pair-wise energy analysis suggested that the energy stabilization from engineered mutations comes not only from electrostatic interactions with E2K, but also from pi-pi stacking interactions with T59F and hydrophobic interactions from A33P as well. The K2$_{HC}$ substitution in Combo1 and mm28 converted this residue to a positive charge and became attractive to the negatively charged E87 in gp120, which reduced the energy 0.76 (Combo1) and 1.09 kcal/mol (mm28) compared to the VRC34.01 template, which tightened and stabilized the binding between the antibody and trimer (Fig. 7D, first column). The T59F$_{HC}$ mutation in Combo1 and mm28 also energetically contributed to stabilization of the antibody itself by generating pi-pi stacking interactions with Y94$_{LC}$ and F59$_{HC}$, thereby reducing the energy 4.96 and 4.55 kcal/mol for Combo1 and mm28, respectively (Fig. 7D, second column). A33P additionally contributed to affinity improvement between the antibody and trimer; the P33$_{HC}$ substitution in mm28 interacted repulsively with I515$_{FP}$ which allowed room for N52$_{HC}$ to generate hydrogen bonding with the carbonyl group in the backbone of I515$_{FP}$ (Fig. 7D, third and fourth columns). The P33$_{HC}$ effect was especially dramatic when comparing mm28 to both VRC34.01 template and Combo1, with the new hydrogen bond resulting in an energy reduction of 5.52 and 3.48 kcal/mol, respectively (Fig. 7D, bottom right graph).

**A** Cryo-EM surface of VRC34.01_mm28 and HIV-1 trimer complex

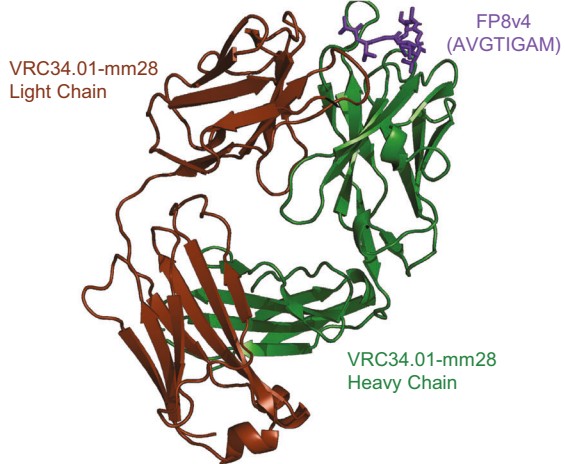

**C** Crystal structure of VRC34.01_mm28 with fusion peptide

**B** Cryo-EM ribbon overlay of VRC34.01 and VRC34.01_mm28 and HIV-1 trimer complex

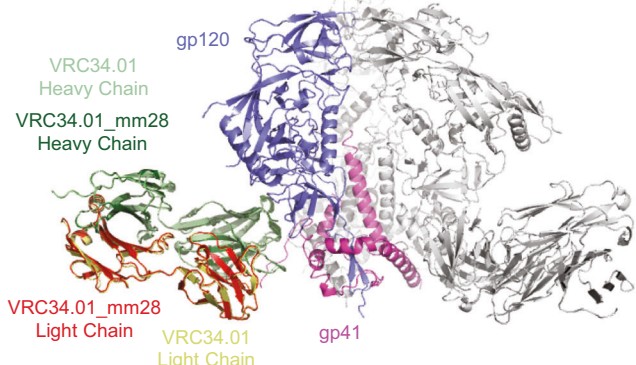

**D** Crystal structure of VRC34.01_mm28 paratope shows A33P location relative to fusion peptide

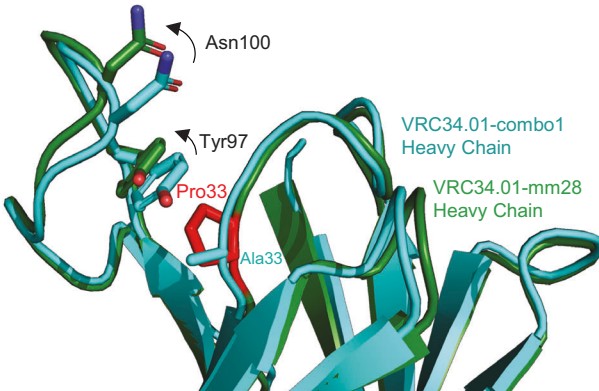

**E** The Ala33Pro mutation in VRC34.01_mm28 generates inter-chain augmentations which displaces Tyr97 and shifts Asn100

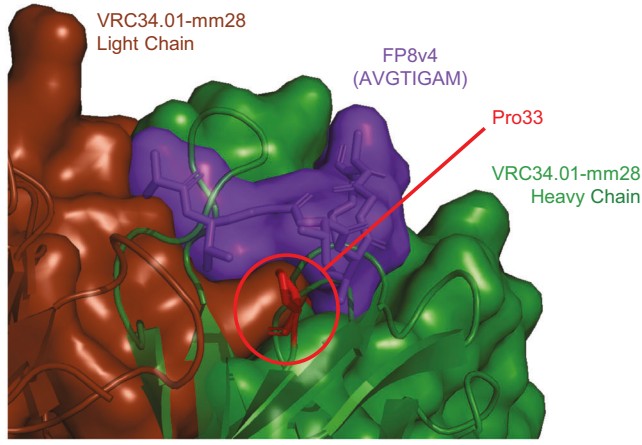

**F** VRC34.01_mm28 has a wider peptide binding cleft than template VRC34.01 by ~1 Å to accommodate diverse FP sequences

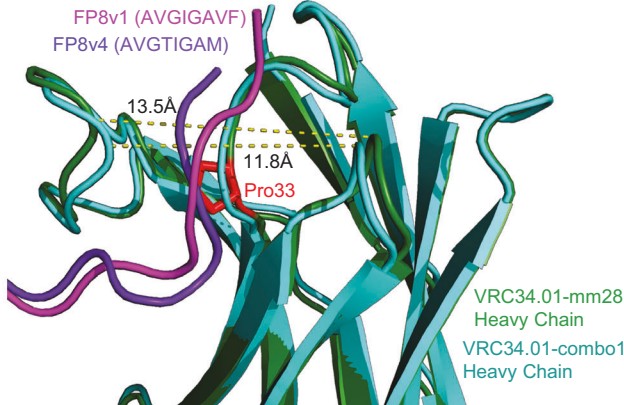

**Fig. 6 | Structural basis for VRC34.01-mm28 recognition of diverse FP.**
**A** CryoEM reconstruction of VRC34.01_mm28 bound to BG505-DSSOSIP. False colors are shown for a single copy each of gp120, gp41 and the heavy and light chains of VRC34.01_mm28. **B** Superposition of VRC34.01 and VRC34.01_mm28 bound to BG505, showing no discernable differences in binding. **C** Crystal structure of VRC34.01_mm28 bound to FP8v4 (AVGTIGAM) with (**D**) an enlarged surface view of the peptide binding region. **E** Mutation of Ala33 to Pro in VRC34.01_mm28 causes a cascading shift in Tyr97 and Asn100, resulting in the enlarged binding cleft. **F** Superimposition of VRC-34.01_Combo1 bound to FP8v1 and VRC34.01-mm28 bound to FP8v4, demonstrating the broader binding cleft of mm28.

## Discussion

Here we revealed a best-in-class antibody against the HIV-1 FP that expands the reported neutralization breadth for an antibody targeting the FP epitope. We conducted an antibody improvement campaign to identify this improved antibody, testing our hypothesis that enhanced VRC34.01 antibody affinity against neutralization-resistant HIV-1 FP residues would lead to improved neutralization breadth. We show that enhanced antibody recognition of neutralization-resistant FP residues aggregated diverse FP recognition and improved neutralization potency and breadth on a globally representative panel of 208 HIV-1

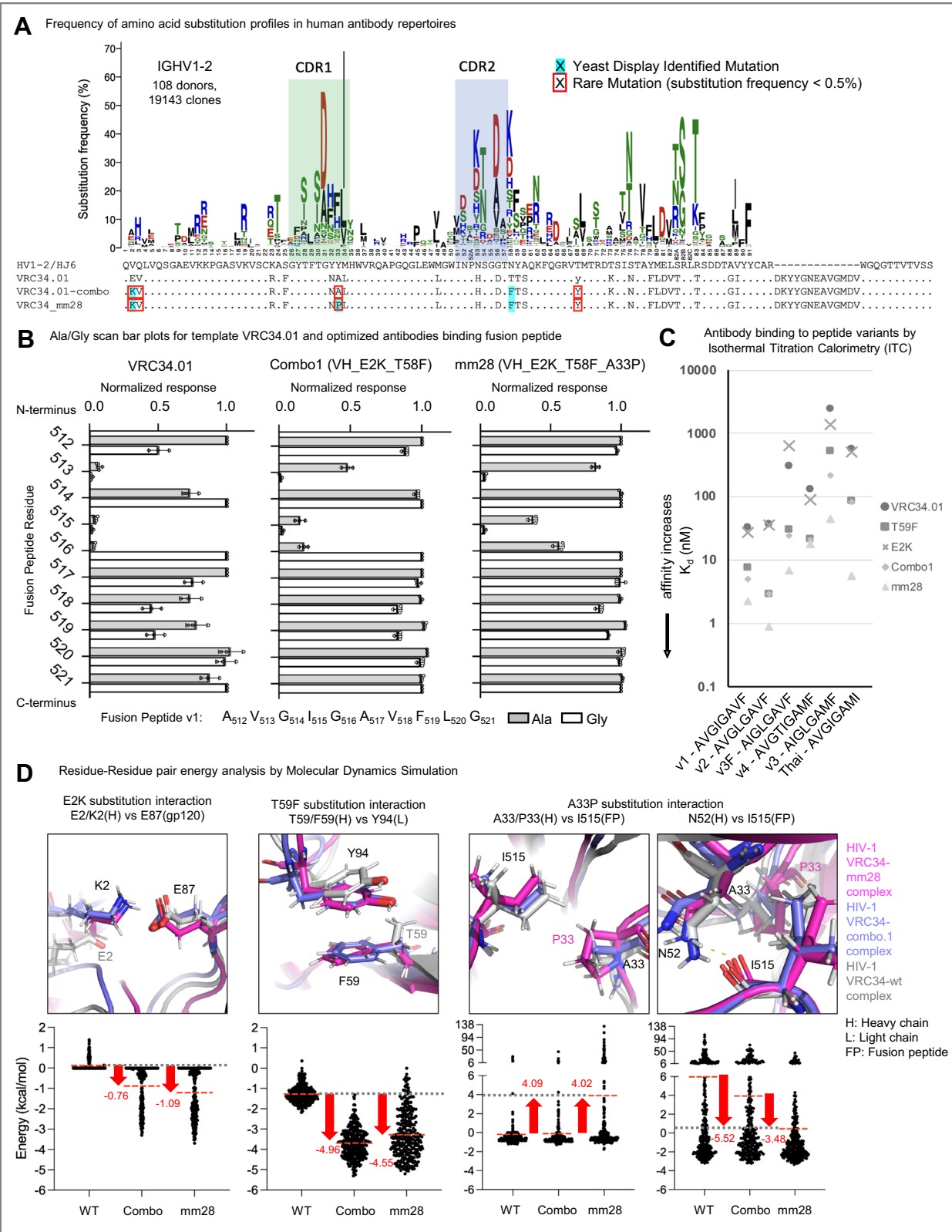

**A** Frequency of amino acid substitution profiles in human antibody repertoires

**B** Ala/Gly scan bar plots for template VRC34.01 and optimized antibodies binding fusion peptide

**C** Antibody binding to peptide variants by Isothermal Titration Calorimetry (ITC)

**D** Residue-Residue pair energy analysis by Molecular Dynamics Simulation

isolates. Our approach substantially improved antibody performance, even for a previously established human bNAb VRC34.01 that had been elicited from chronic infection and exposure to substantial natural viral diversity[26]. The most improved antibody, VRC34.01_mm28, comprised three amnio acid substitutions from VRC34.01 in the heavy chain variable region and achieved best-in-class neutralization with 10-fold enhanced potency and ~80% breadth on a 208-isolate panel of

circulating HIV-1 strains, a significant increase in breadth by ~30% compared to the template antibody VRC34.01[26]. In parallel, we defined a panel of anti-FP mAbs and their functional antibody:antigen interactions for improved FP-based recognition of HIV-1.

Our efforts complement other recent data in the field to indicate the high promise of targeting FP for HIV-1 vaccine designs[26–31]. A major goal of HIV-1 vaccine development is to elicit broadly neutralizing

**Fig. 7 | Biophysical and genetic correlates for best-in-class FP-directed antibody neutralization. A** Rare mutations identified through gene-specific substitution profiles (GSSPs) critical for the recognition of FP and HIV−1 Env trimer. GSSPs for the VRC34.01_Combo1 heavy chain *IGHV1-2* gene are shown. Mutations identified in yeast display were highlighted with cyan background. A substitution frequency <0.5% was defined as a rare mutation and colored with a red square. *CDR1* and *CDR2* were shown with green and blue boxes, respectively. **B** Binding of optimized antibodies to His-tagged FP Ala/Gly mutants. Binding of alanine (gray bars) and glycine (white bars) mutants within FP, normalized by binding to the wild-type

sequence, are shown, with FP amino acids on the *y*-axis. Experiments were performed in triplicate (*n* = 3 independent experiments), error bars indicate the mean with SD plotted for each residue position in the overlaid scatter dot plots. **C** ITC-derived affinity to diverse FP sequences for various antibody variants. **D** Residue-Residue Pair Energy Analysis by 100 ns Molecular Dynamics Simulation to understand the roles of E2K, T59F, and A33P mutations in improved neutralization performance. Mean is represented as red dashed lines, and red arrows show the energy difference between mutants and the VRC34.01 (WT) antibody. Gray dashed lines represent VRC34.01 (WT) performance for comparison.

antibody (bNAb) responses and the FP epitope represents a promising vaccine target; the linear eight amino acid target sequence presents favorable features for focused immunogen design because it is linked to the critical viral entry function, which is common among Class I viral fusion proteins. Additionally, conserved FP features can be recognized by antibodies elicited from both natural infection and vaccination[26–29]. The N-terminus of the fusion peptide on HIV-1 Env is accessible when exposed in the closed, prefusion state, and several of these hydrophobic residues can enable high-affinity binding interactions with the antibody paratope[30,31]. Our scientific understanding of effective bNAb development pathways targeting FP has significantly advanced in the past decades, fueling rational vaccine designs and therapeutic drug development against HIV-1. While FP-directed vaccines have shown increasing promise in multiple animal models several outstanding challenges remain for broad HIV-1 immunization in humans, and a better understanding of how to attain sufficient breadth and potency against the FP epitope is essential for further progress[28,29].

Several molecular mechanisms constrain HIV-1 FP recognition and impede in vivo bNAb development. Prior anti-FP antibody structures demonstrated that viral mutations in FP and in distal sites of Env can mediate viral escape from both natural- and vaccine-elicited antibodies, and we observed some resistant mutations in FP for VRC34_mm28 (Supplemental Table 9)[26–31]. In parallel to viral escape, the accumulation of antibody mutations during in vivo development is naturally limited by statistical sampling bias related to established SHM base preferences[33,35]. Antibody mutations that require multiple nucleotide substitutions per amino acid codon may not be readily sampled by in vivo affinity maturation, however, these rare mutations are critical for HIV-1 bNAb development[42,43]. To overcome natural mutational sampling constraints, here we comprehensively screened single mutant substitutions using yeast display, revealing that improvements in protective breadth can be achieved with combinations of very few, albeit infrequently sampled, sets of mutations to the template antibody sequence. Our findings support previous observations in HIV-1 bNAb development that rare mutations can often provide remarkable improvements to antibody function, although such rare mutations may be difficult to sample in vivo.

Structural comparisons of the template VRC34.01 to the top two engineered variants VRC34.01_Combo1 and VRC34.01_mm28 demonstrated that improved antibodies still bound to FP and Env much like the template antibody, relying primarily on heavy chain interactions with both gp120 and gp41 for virus recognition, with improved antibodies binding more surface area at the FP interface. Structural & biophysical analysis provided further insights into neutralization improvement mechanisms, with beneficial mutations stabilizing residue pair energies and enabling paratope to accommodate for insertions and substitutions in the FP sequence, resulting in enhanced antibody affinity to neutralization-resistant FP sequences for improved neutralization potency and breadth. We aligned FP sequences from the 208-virus panel and showed the number of sensitive and resistant FP sequences to VRC34.01_mm28 in Supplemental Table 9 (left). Sensitive and resistant strains are shown in Supplemental Table 9 (right), with specific positions and amino acids associated with resistance shown in red. Recognition of diverse FP sequences improved with the

substitutions comprising VRC34.01_mm28, and our analysis of resistant amino acids at different positions shows several resistant amino acid residues that could be used to further engineer anti-FP antibodies like mm28 for even greater neutralization breadth.

Biophysical characterization of sequence-structure-function relationships enhanced our understanding of the molecular interactions required for broad FP recognition[44]. The three mutations in VH_E2K_A33P_T59F comprising the most improved variant VRC34.01_mm28 worked synergistically to enhance neutralization by providing favorable binding dynamics that increased broad HIV-1 trimer recognition. The rare VH_E2K substitution modified interchain electrostatic interactions, while VH_T59F influenced pi-pi interactions to increase favorable antibody interactions with FP, and A33P widened the FP binding groove to accommodate sequence diversity in circulating strains[26]. The same A33P mutation was previously observed in VRC34.04, but without a similar overall improvement in performance[43], and the three-point mutations described here must be combined into a single molecule for their full synergistic effects.

The data presented here provide a molecular blueprint for broad and potent anti-FP targeting using human antibodies as guiding templates. VRC34.01_mm28 is an anti-FP mAb with neutralization breadth on par with traditionally broad-recognition epitopes like the CD4 binding site[6,45–47], lending further promise to the FP epitope for targeted vaccine design. Our findings support immunization strategies using both 8-mer and 9-mer FP variants to elicit antibodies that can recognize globally diverse HIV-1 strains. An ideal vaccine strategy may be targeted elicitation of a single broad and potent antibody lineage resulting in humoral immunity against both FP variant lengths, like the A33P mutation described here to broaden the FP binding site. However, the elicitation of separate sets of antibodies targeting 8-mer and 9-mer FPs could also be plausible through other antibody maturation pathways, and may be sufficient to achieve high neutralization breadth and potency via vaccination.

Interestingly, our results showed that while the template antibody light chain was sensitive to mutant variations, none of the light chain mutations showed substantial benefit compared to the heavy chain mutations alone in mm28 (Supplemental Fig. 4B). Notably, rationally designed multi-mutation variant mm28-VL_R30H_S92T, containing two additional light chain mutations in addition to mm28 mutations, achieved similar neutralization efficiency as mm28 on a 20-virus neutralization panel, and may be an interesting candidate for follow-up analysis to better understand light chain interaction mechanisms for FP mAbs.

Further potency improvements from bNAbs like VRC34.01_mm28 will support ongoing translational efforts developing and administering prophylactic medicines to prevent HIV-1 transmission. Antibody improvement campaigns and their associate structural & biophysical insights are important tools to accelerate structure-guided interventions against infectious diseases[7,33–35]. Our results outlined the molecular features of truly broad HIV-1 neutralization targeting the FP epitope, and the general methods and approach can also be applied more generally to produce highly functional antibodies against other clinically relevant pathogens like influenza virus, Ebola virus, and malaria[33,34,40,48]. We anticipate that future efforts will build on the data

presented in this study to address clinically relevant infectious diseases and establish potent, near-universally protective FP interventions to suppress the ongoing HIV-1 pandemic.

## Methods

### SSM library generation and cloning into yeast display

One-pot site saturation mutagenesis (SSM) protocol single-mutation plasmid libraries were constructed as previously described[33,34,39] (Supplemental Table 1). VRC34.01 heavy chain variable region (*VH*) and light chain variable region (*VL*) template genes were mutated separately using mutagenic primers containing degenerate single codons (NNK or MNN) to express all 20 amino acids at each residue of the *VH* and *VL*. A process of single-strand nicking, exonuclease digestion, and degenerate oligonucleotide-primed PCR was used to apply the mutagenic primers to the template genes and generate each library (*VH* and *VL*) containing the comprehensive set of single-residue substitutions across the VRC34.01 variable regions[49]. Using high-efficiency electrocompetent E. coli (New England Biolabs, Cat #C2987H) separate $VH_{SSM}$:$VL_{template}$ and $VH_{template}$:$VL_{SSM}$ libraries were cloned into a yeast display plasmid vector containing a FLAG-marker to quantify Fab surface expression, as well as a leucine zipper and protein disulfide isomerase expression for enhanced expression of diverse antibody libraries[40]. Plasmid DNA libraries were used to transform AWY101 (MATa AGA1::GAL1-AGA1::URA3 PDI1::GAPDH-PDI1::LEU2 ura3-52 trp1 leu2 Delta 1 his3 Delta 200 pep4::HIS3 prb1 Delta 1.6 R can1 GAL; Eric Shusta Lab, University of Wisconsin) yeast populations, and library sizes of at least $2 \times 10^6$ were maintained in all cloning steps, as described previously[33].

### Fluorescence-activated cell sorting of single-mutation libraries

Transformed yeast libraries were cultured in SGDCAA medium (20 g/L galactose supplemented with 2 g/L dextrose, 6.7 g/L yeast nitrogen base, 5 g/L casamino acids, 5.4 g/L $Na_2HPO_4$, 8.6 g/L $NaH_2PO_4.H_2O$; SGCAA from TEKnova, Hollister, CA) for 36 h at 20 °C and 225 rpm to induce antibody fragment (Fab) surface expression. Induced yeast libraries were washed and stained with an anti-FLAG FITC monoclonal to quantify Fab expression (F4049, Clone M2, 1:49 dilution, Sigma-Aldrich, Burlington, MA). Trimer antigen probes were generated by fluorescently labeling biotinylated constructs with an anti-biotin-PE label (12-9895-82, clone BK-1/39, 1:4 dilution, Thermo Scientific, Waltham, MA). Anti-FLAG-FITC-labeled libraries were co-stained with fluorescently conjugated antigen probes to screen single mutation Fab libraries for antigen recognition. In the first round of FACS, a minimum of 3e7 yeast cells were stained and sorted using three gates to sort and collect low-, medium-, and high-affinity yeast library groups measured by the ratio of Fab surface-expression versus Ag binding for the cell population, as described previously[33,48]. Sequential gating examples for all flow cytometry plots are provided in Fig. S1A of ref. 33. For all sorting experiments, control libraries of FITC+ yeast were also collected to track the identity and initial prevalence of clonal variants with successful Fab expression. Collected cells were grown in low-pH SDCAA (20 g/L dextrose, 6.7 g/L yeast nitrogen base, 5 g/L casamino acids, 10.4 g/L trisodium citrate, and 7.4 g/L citric acid monohydrate, pH 4.5) for 24-48 h in a 30 °C incubator shaking at 225 rpm post-sorting. After sorted yeast collection and culture, each screening sort was repeated (using either low-, medium-, or high-affinity gates to match the initial Round 1 sort) for an additional two rounds.

### Multi-mutation library design and screening

Plasmid libraries were isolated from sorted yeast cells using previously described DNA extraction methods[49]. *VH* and *VL* genes were amplified from library plasmids using Kapa Hifi HotStart ReadyMix (Kapa Biosystems, Roche. Wilmington, MA)[40]. Five paired *VH*:*VL* multi-mutation libraries were designed using template DNA from the enriched single-mutation high-affinity screens (Supplemental Table 1). Library 1 ($VH_{Shuffled}$:$VL_{template}$) and Library 2 ($VH_{template}$:$VL_{Shuffled}$) were generated by shuffling enriched single mutations on VH and VL genes, respectively, using a previously defined protocol[50]. Template DNA was fragmented with DNAseI, followed by homologous reassembly and reamplification of the shuffled genetic material. Library 3 was generated by pooling FACS-enriched single-mutation *VH* and single-mutation *VL* gene libraries and then sub-cloning into a shared expression vector via restriction enzyme digest to randomly combine single mutations from both heavy and light chains. Library 4 was created by performing DNA shuffling on Library 3, and Library 5 was generated by repeating another round of SSM on Library 4.

Multi-mutation plasmid libraries were transformed into yeast libraries using electroporation, as with single mutation library screens above, with library sizes exceeding $3 \times 10^6$ maintained in all cloning steps. Expression was induced and yeast were stained as above using an anti-FLAG FITC monoclonal antibody (F4049, Clone M2, Sigma-Aldrich. Burlington, MA) and a fluorescently anti-biotin-PE labeled antigen probe (12-9895-82, clone BK-1/39, Thermo Scientific. Waltham, MA). Control libraries of FITC+ yeast from the pre-sort libraries were also collected to track the identity and initial prevalence of multi-mutation variants with successful Fab expression. For screening sorts, libraries were individually stained with 0.3 nM FP_v1, 0.7 nM FP_Thai, 0.7 nM FP_v3, and 7 nm FP_v4 and enriched consistently against the same antigen across multiple rounds. Multi-mutation libraries were enriched only for high-affinity binding. An excess of $3 \times 10^7$ yeast cells were stained and the 0.1% highest affinity variants, defined by the highest ratio of antigen binding to Fab expression, were gated and collected for an additional three rounds using FACS[33,48]. Collected cells were cultured in low-pH SDCAA (20 g/L dextrose, 6.7 g/L yeast nitrogen base, 5 g/L casamino acids, 10.4 g/L trisodium citrate, and 7.4 g/L citric acid monohydrate, pH 4.5) for 24-48 h at 30 °C and 225 rpm. Flow cytometry results were analyzed using FlowJo™ v10.8 Software (BD Life Sciences).

### NGS and bioinformatic analysis of sorted antibody libraries

*VH* and *VL* genes were extracted and amplified from sorted yeast library cultures and submitted for Next Generation Sequencing (NGS) on an 2 × 300 bp Illumina MiSeq platform. First, plasmids were isolated from cultured yeast cells as described previously[49] and then *VH* and *VL* genes were amplified from library plasmids using Kapa Hifi HotStart ReadyMix (Kapa Biosystems, Roche. Wilmington, MA) as described previously[33,40,51]. Amplified libraries were prepared for NGS using an addition round of PCR to incorporate barcodes and adapters for Illumina sequencing.

Raw Illumina fastq sequence reads were processed as described previously[33,51,52]. NGS reads were quality-filtered for a score of 30 over 90% of the raw reads using Fastxtoolkit (v0.0.14 http://hannonlab.cshl.edu/fastx_toolkit/). Filtered reads were processed using IgBlast software to reference the IMGT database and determine complete variable region gene alignments[53,54]. Once variable region alignments were obtained, mutant VRC34.01 sequences were aligned to the template VRC34.01 antibody sequence using Usearch[55]. Mutations were defined by determining the percent identity match to the template gene and denoting the substituted amino acid residue according to the template sequence, with Base 1 corresponding to the start of the variable framework region. Kabat numbering identifiers are also listed for key mutations and referred to in structural data (Supplemental Table 2). The number of reads for unique sequences were enumerated in each

library, and used to determine the prevalence, or frequency, of each variant in the sorted libraries:

$$Prevalence_{Variant\,X\,in\,Sorted\,Library\,Y} = \frac{Number\,of\,Reads\,of\,Sequence\,X\,in\,Sorted\,Library\,Y}{Total\,Number\,of\,Reads\,in\,Sorted\,Library\,Y} \tag{1}$$

We defined a variant's enrichment ratio (ER) as the change in sequence prevalence from the initial Fab expressing (VL-FITC+) control library collected without antigen to the experimentally Ag-screened library:

$$ER_{Sequence\,X\,in\,Sorted\,Library\,Y} = \frac{Prevalence\,of\,Sequence\,X\,in\,Sorted\,Library\,Y}{Prevalence\,of\,Sequence\,X\,in\,Fab\,Expressing\,Pre-Sort\,Control\,Library} \tag{2}$$

Each single mutation variant was binned into a high-, medium-, or low-affinity population by comparing prevalence and enrichment ratio values across different screening conditions, as we reported previously[33,48]. Multi-mutation variants were analyzed by their ER in high-affinity sorted library screens, using the multi-mutation VL-FITC+ prevalence in the denominator, and also by the prevalence of multi-mutation variants in the final round of sorting.

### Antibody expression
Antibody variable region heavy chain and light chain sequences were codon optimized, synthesized, and cloned into a VRC8400 (CMV/R expression vector)-based IgG1 vector as previously described[29]. Antibodies were expressed by transient transfection in Expi293 cells (ThermoFisher Scientific, Waltham, MA) using Turbo293 transfection reagent (SPEED BioSystems, Gaithersburg, MD) according to the manufacturer's recommendation. 50 μg plasmid encoding heavy-chain and 50 μg plasmid encoding light-chain variant genes were mixed with the transfection reagents, added to 100 ml of cells at $2.5 \times 10^6$/ml, and incubated in a shaker incubator at 120 rpm, 37 °C, 9% CO2. At 5 days post-transfection, cell culture supernatant was harvested and purified with a Protein A column (GE Healthcare, Chicago, IL). The antibody was eluted using IgG Elution Buffer (ThermoFisher Scientific, Waltham, MA) and brought to neutral pH with 1 M Tris-HCl, pH 8.0. Eluted antibodies were dialyzed against PBS overnight and were confirmed by SDS-PAGE before use.

### Virus neutralization assays
**8- and 20-virus panels.** Monoclonal antibodies were assessed one-on-one against multiple viral strains using entry neutralization assays as previously described neutralization[6]. Five-folding serial dilutions of antibodies starting at 500 μg/mL were mixed in 50 μL volumes with stocks of viruses carrying fluorescent luciferase reporter genes. Mixtures were incubated at 37 °C for 1 h, followed by the addition of 20 μL TZM-bI cells ($0.5 \times 10^6$ cells/mL, NIH AIDS Reagent Program, Bethesda, MD) and incubation overnight at 37 °C. After 24 h (day 2) an additional 130 μL of complete Dulbecco's modified Eagle medium was added to the neutralization test reactions and incubated at 37 °C overnight. On day 3 cells were lysed and assessed for luciferase activity indicative of viral infection by measuring in relative light units. The concentration of antibody required to inhibit 50 and 80% of virus infection as determined by comparing relative light units between samples to negative non-neutralized controls were determined using a hill-slope regression analysis as described[6,33].

**208-virus panels.** To model monoclonal antibody function against globally circulating fusion peptide diversity an automated large-batch neutralization panel of 208 HIV-1 Env-pseudotyped viruses was performed using 384-well microneutralization assays as described previously[56].

### Improved VRC34 mAb crystal structure determination
Fabs of VRC34-combo1 and VRC34-mm28 were produced by proteolytic cleavage of IgGs of the antibodies containing an engineered HRV-3C cleavage site at the hinge region. Cleaved Fabs were applied to a Superdex S-75 gel filtration column and equilibrated with 5 mM HEPES pH 7.5, 150 mM NaCl. Fabs were concentrated to 14 mg/ml and mixed with either FP8v1 (VRC34_Combo1) or FP8v4 (VRC34_mm28) peptides at a molar ratio of 2:1. These mixtures were crystallized by hanging drop vapor diffusion over a well solution of 40% MPD and 0.1 mM sodium potassium phosphate pH 6.2 in the case of VRC34_combo1, or 10% PEG 3,350 and 1.4 M sodium/potassium phosphate pH 7.5 for VRC34_mm28. In both cases, crystals were diffracted at the ID-22 beamline at the Advanced Photon Source (Argonne National Laboratory, Lemont, IL), and data reduction performed with HKL3000. Molecular replacement was performed using the VRC34.01 structure (PDB ID: 5I8E), after which the structures were built and refined using Coot and the Phenix package, and figures generated using Pymol.

### Cryo-EM structure determination
In all, 6.5 mg/ml BG505-DS-SOSIP was mixed with 8 mg/ml Fabs at molar ratio of 1:1.2. Quantifoil R 2/2 gold grids were glow-discharged with PELCO easiGlow glow-discharger (0.39 mBar, 20 mA, and 30 s). Cryo-EM grids were prepared on FEI Vitrobot Mark IV plunger at the chamber temperature of 4 °C and humidity of 95%; the sample volume was 2.7 μl. Cryo-EM datasets were collected at the National Cryo-Electron Microscopy Facility (NICE, Frederick, MD) on an FEI Titan Krios electron microscope equipped with a Gatan K2 summit DED operated in the super-resolution mode (pixel size before binning: 0.415 Å) (Supplemental Table 7). Reconstruction of HIV-Fab complexes was carried out with cryoSPARC v3.3[57]. The movies were aligned and dose-weighted using patch motion correction. The micrograph contrast transfer function (CTF) parameters were found using patch CTF estimator. Particles were picked with the blob picker and subjected to 2D classification with selection of best classes. Ab-initio reconstruction and non-uniform refinement were run with imposed C1 and C3 symmetry, respectively. To obtain initial atomic models, the structures of BG505-DS-SOSIP (PDB 6V0R) and VRC34.01 Fab (PDB 5I8E) were docked into corresponding cryo-EM maps in UCSF Chimera[58]. E2K, T59F/T58F, and A33P substitutions were made in Coot[59,60]. Atomic models were refined by alternating rounds of model optimization in Coot and real-space refinement in Phenix[61]. Structure validation was performed with Molprobity and the PDB validation server[62]. Summaries of model refinement statistics and quality assessment for cryo-EM reconstructions are given in Supplemental Table 7, and in Supplemental Figs. 3, 5, and 6.

## Isothermal titration calorimetry analysis

The binding of VRC34.01 and the four variants T59F, E2K, Combo1, and mm28 to DS-SOSIP and six different fusion peptides were measured by ITC:

V1: AVGIGAVFLGGGKKKGGHHHHHHHH;
V2: AVGLGAVFLGGGKKKGGHHHHHHHH;
V3: AIGLGAVFLGGGKKKGGHHHHHHHH;
V4: AVGTIGAMFLGGGKKKGGHHHHHHHH;
V3M: AIGLGAMFLGGGKKKGGHHHHHHHH;
V5Thai: AVGIGAMILGGGKKKGGHHHHHHHH.

ITC experiments were performed at 25 °C using a VP-ITC from MicroCal/Malvern Instruments (Northampton, MA, USA). DS-SOSIP and the antibodies were dissolved in PBS, pH 7.4, and thoroughly dialyzed prior to the experiments. The dialysate was used to dissolve the different peptides. In each titration, the antibody was added stepwise in 7-μL aliquots to the stirred calorimetric cell ($v \sim 1.4$ mL) containing either peptide at 1 μM or DS-SOSIP at 0.5–0.7 μM trimer. The antibody concentration in the syringe was 11–15 μM antigen binding sites. The concentrations of antibody and DS-SOSIP were determined from the absorbance at 280 nm while the concentration of each peptide was obtained from the total nitrogen concentration determined according to Jaenicke[63]. The heat produced upon each injection was obtained from the integral of the calorimetric signal, and the heat associated with binding was obtained after subtraction of the heat of dilution. The association constant, $K_a$ (the dissociation constant, $K_d = 1/K_a$), the enthalpy change, $\Delta H$, and the stoichiometry, $N$, were obtained by nonlinear regression of the data to a single-site binding model. Gibbs energy change, $\Delta G$, was obtained from the binding affinity using $\Delta G = -RT \ln K_a$, ($R = 1.987$ cal $\times$ K$^{-1}$ $\times$ mol$^{-1}$ and $T$ is the absolute temperature in kelvin). The entropy contribution to Gibbs energy change, $-T\Delta S$, was calculated from the relation $\Delta G = \Delta H - T\Delta S$. All the results are expressed per mole of antigen binding sites. The stoichiometry, $N$, denotes the number of antigen-binding sites per peptide and DS-SOSIP trimer, respectively.

## Alanine/glycine scanning analysis

Binding of WT and optimized VRC34 antibodies to sixteen different His-tagged fusion peptides (residue 512-521), including wild type and alanine/glycine mutants, was assessed using a fortéBio Octet HTX instrument. His-tagged fusion peptides were synthesized (GenScript) with an eight-residue ggKKKggg linker followed by an eight-histidine residue tag at the C terminus of FP. Briefly, the sixteen peptides at 50 μg/ml in PBS were loaded onto Ni-NTA biosensors using their C-terminal histidine tags for 60 s. Typical capture levels were between 1.3 and 1.5 nm and variability within a row of eight tips did not exceed 0.1 nm. These peptide-bound biosensors were equilibrated in HBS-P+ containing 50 μM EDTA buffer (HBS-EP +) for 60 s followed by capture of the antigen binding fragments (Fabs, 250 nM in HBS-EP+) of the WT and optimized VRC34 antibodies, and an RSV F antibody Motavizumab was included as a negative control, for 120 s and a subsequent dissociation step in HBS-EP+ buffer for 300 s. Parallel correction to subtract systematic baseline drift was carried out for all sixteen peptides by subtracting the measurements recorded for each individual peptide loaded sensor incubated in HBS-EP+. Data analysis was carried out using Octet software, version 12.0.2. The normalized responses obtained from independent triplicate datasets were plotted using PRISM (PRISM 9 GraphPad Software).

## Molecular dynamic model simulations

Initial crystal structures PDB: 5I8H, 8ELI, and 8F7Z were used for VRC34.01 WT, Combo1, and mm28 respectively, and missing residues were modeled by YASARA[64,65] using them as corresponding template structures. Man-5 glycosylation in potential $N$-linked glycan sites was prepared by Glycan Reader & Modeler[66] in CHARMM-GUI[67].

Molecular dynamics (MD) simulations were carried out for three complex structures – VRC34.01 WT, Combo1, mm28 bound to Man-5 glycosylated complexes. The systems were solvated in water boxes with explicit water model TIP3P[68] and neutralized by the addition of potassium and chloride ions at a concentration of 0.15 M. The final systems were composed of 834,375 (WT), 822,734 (Combo1), 822,827 (mm28) atoms. All atom MD simulations were performed using NAMD2.13[69], with CHARMM36 force field[70,71]. The simulations were conducted using periodic boundary conditions and particle-mesh Ewald (PME) electrostatics summation[72] with maximum grid spacing 1 Å. The van der Waals interaction cutoff was 12 Å, with a switching distance 10 Å. The system was first minimized 10,000 steps by conjugate gradient method, and then equilibrated 125,000 steps using a linear temperature gradient, which heated up the system to 310 K. All simulations were performed with NPT ensemble (310 K 1.01325 bar) with Langevin thermostat[73] and Nosé-Hoover Langevin pressure control[74–76]. The production step was carried out for 100 ns with 2 fs/step and used in energy analysis.

## Residue-residue pair energy analysis

Rosetta Interface Analyzer[77,78] was used to analyze residue-residue pair energy. The interface residue pairs HIV-1 trimer vs VRC34 lineage antibodies (WT, Combo1, mm28), and VRC34 heavy vs. light chains were defined. Only the residue pairs within 12 Å distance cutoff were included in the calculation.

## Data availability

Antibody library screening and characterization data are available in the manuscript files. Source data are provided with this paper as Supplementary Files. Antibody sequence data are deposited at NCBI Genbank, accession codes OR327479-OR327564. Structural data are deposited online in the PDB under accession codes PDB: 8EUU, PDB: 8EUV, and PDB: 8EUW.

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

## Acknowledgements

We thank J. Hackett for help with data collection. We thank J. Baalwa, D. Ellenberger, F. Gao, B. Hahn, K. Hong, J. Kim, F. McCutchan, D. Montefiori, L. Morris, J. Overbaugh, E. Sanders-Buell, G. Shaw, R. Swanstrom, M. Thomson, S. Tovanabutra, C. Williamson and L. Zhang for contributing the HIV-1 Envelope plasmids used in our neutralization panel. We thank R. Carroll, N. Jean-Baptiste, C. Moore, S. O'Dell, S.D. Schmidt, C. Whittaker, and A.B. McDermott for their assistance with neutralization assessments on the 208-strain panel. We thank A. Zeher and R. Huang at the NCI/NICE Cryo-EM Facility for cryo-EM data collection. This work utilized the NCI/NICE Cryo-EM Facility and the computational resources of the NIAID Locus computer center (https://locus.niaid.nih.gov). Use of sector 22 (Southeast Region Collaborative Access team) at the Advanced Photon Source was supported by the US Department of Energy, Basic Energy Sciences, Office of Science, under contract no. W-31-109-Eng-38. This work was supported by a University of Kansas Self Graduate Fellowship (B.B.B.) and the University of Kansas Departments of Chemical Engineering and Pharmaceutical Chemistry, the National Cancer Institute (HHSN261200800001E to A.S.), and by NIH grants DP5OD023118, P20GM103418, R01AI141452, and R21AI143407, and U01AI169587 (to B.J.D), and by the Intramural Research Program of the Vaccine Research Center, National Institute of Allergy and Infectious Diseases, NIH.

## Author contributions

Designed experiments: B.B.B., S.P., A.S.O., K.X., B.Z., R.R., T.B., N.A.D-R., T.D.N., G-Y.C., J.R.M., P.D.K., and B.J.D.; Performed experiments: B.B.B., S.P., A.S.O., K.X., B.Z., T.B., N.A.D-R., T.D.N., M.L., B.C.L., T.L., M.L., K.M., S.O., M.S., A.S., N.B., and J.R.W.; Analyzed the data: B.B.B., S.P., A.S.O., K.X., B.Z., R.R., T.B., N.A.D-R., A.S.F., M.Lee., B.M., K.M., S.O., M.S., A.S., C-H.S., P.D.K., and B.J.D.; Writing: B.B.B., P.D.K, and B.J.D.; Reviewing and editing: all authors.

## Competing interests

The authors declare no competing interests.
