## [Peer review file · Nature Communications]

REVIEWER COMMENTS

Reviewer #1 (Remarks to the Author):

This paper describes the in vitro engineering of an antibody binding the 8-9 residue HIV fusion peptide (FP) from the initial VRC34.01 clone, originally isolated from a chronically-infected HIV-1 donor. The initial clone neutralizes 50% of a panel of 208 strains at 50 ug/ml to a very broadly neutralizing antibody able to neutralize 80% of these strains panel with up to 10-fold increased potency for some strains. This was achieved using a very systematic approach this group has previously applied to other variable antigens; accordingly, this report demonstrates the generality of their approach. Single-site mutagenesis libraries were first developed and then sorted for low, medium and high binding to HIV envelope proteins containing four diverse FPs. High through put sequencing identified residue changes enriched in high affinity clones but de-enriched in low and medium affinity clones for re-screening in neutralization assays before rationally combining single variants for re-screening. Since this combo1 showed promise, enriched single-site mutation libraries were pooled, shuffled and subjected to single-site mutagenesis again to generate 5 libraries which were screened with four envelope antigens to again sequence sorted clones and generate rationally recombined variants resulting in VRC34.01_mm28 with three heavy chain changes relative to VRC34.01. The biochemical basis of broad reactivity for the final lead variant, VRC34.01mm28, was thoroughly described using a combination of crystallography of the Fab-peptide complex, cryoEM of the Fab-HIV envelope complex, peptide mutagenesis, calorimetry and molecular dynamics simulations.

This report represents a tremendous antibody engineering and characterization effort and is important for several reasons. Generation of mm28 supports the feasibility of very broadly neutralizing antibodies binding the FP and further validates interest in the FP as a target. Most interestingly, however, is the realization that antibodies like mm28 may be hard to elicit by immunization since several of the three four rare changes are required relative to the germline sequence (versus the two rare changes required to elicit VRC34.01). While this report represents just one path from VRC34.01 to a more potent antibody, it also highlights some of the limitations of the FP as an immunogen. Overall, this work shares important results, is well written and should be of interest to Nat Commun readers.

Major comments

- Brief comments on how the parent antibody parses binding to the gp41/ 120 trimer versus the FP in the introduction would set the stage for the later results (eg, what fraction of the Fab buried surface are interacts with FP versus other envelope sequences?). Are different FP sequences more or less accessible which can impact antibody binding?
- The ITC measurements are noted as not providing actual Kds; I am curious as to how to interpret this and the series of Kd values noted that were collected from the ITC experiments (Kds of 33 nM to 2500 nM). If these are not useful Kds, it would be helpful to provide some Kd values (SPR or similar; BLI off-rates may be tricky with the trimer) in order to put the improved binding in context as well as the breadth.

Minor comments:

- Fig 7: the letters in panel A are laterally compressed and hard to read. There is space to stretch it horizontally which I think will help.
- The first stage in generating variants by "precision mutation library generation" is clear and yields an improved clone, but the subsequent steps used to generate mm28 are far less clear in the results. It may help to refer the reader to the methods section.

Reviewer #2 (Remarks to the Author):

Banach and colleagues report on the directed evolution of the anti-fusion peptide antibody VRC34.01 to improve its breadth and potency. The authors argue that defining how this antibody may acquire more broad and potent neutralization will define the features necessary for vaccine